# The Effects of an Intensive Rehabilitation Program on the Nutritional and Functional Status of Post-COVID-19 Pneumonia Patients

**DOI:** 10.3390/nu14122501

**Published:** 2022-06-16

**Authors:** Diogo Sousa-Catita, Catarina Godinho, Paulo Mascarenhas, Filipa Quaresma, Jorge Fonseca

**Affiliations:** 1Grupo de Patologia Médica, Nutrição e Exercício Clínico (PaMNEC) do Centro de Investigação Interdisciplinar Egas Moniz (CiiEM), 2829-511 Almada, Portugal; cgcgodinho@gmail.com (C.G.); pmascarenhas@egasmoniz.edu.pt (P.M.); jorgedafonseca@hotmail.com (J.F.); 2GENE—Artificial Feeding Team, Gastroenterology Department, Hospital Garcia de Orta, 2805-267 Almada, Portugal; 3Residências Montepio—Serviços de Saúde, SA, 1600-131 Lisboa, Portugal; filipa.quaresma@residenciasmontepio.pt

**Keywords:** COVID-19 pneumonia, nutritional and functional status, interdisciplinary rehabilitation program

## Abstract

Most hospitalized COVID-19 pneumonia patients are older adults and/or have nutrition-related issues. Many are bedridden in intensive care units (ICU), a well-documented cause of malnutrition, muscle wasting, and functional impairment. Objectives: To assess the effectiveness of an intensive rehabilitation program over the nutritional/functional status of patients recovering from COVID-19 pneumonia. Post-COVID-19 pneumonia patients underwent a 30-day intensive interdisciplinary rehabilitation program including a personalized nutritional intervention designed to achieve a minimum intake of 30 kcal/kg/day and 1 g protein/kg/day. The nutritional and functional status was assessed in each patient at three different moments. Each assessment included Body Mass Index (BMI), Mid Upper Arm Circumference (MUAC), Mid Arm Muscle Circumference (MAMC), Tricipital Skinfold (TSF), Hand Grip Strength (HGS), and Mini Nutritional Assessment (MNA^®^). The study included 118 patients, with ages in the range 41–90 years old. BMI increased linearly over time (0.642 units, F-test = 26.458, *p* < 0.001). MUAC (0.322 units, F-test = 0.515, *p* = 0.474) and MAMC status (F-test = 1.089, *p* = 0.299) improved slightly, whereas TSF decreased (F-test = 1.885, *p* = 0.172), but all these arm anthropometry trends did not show significant variations, while HGS (4.131 units, F-test = 82.540, *p* < 0.001) and MNA^®^ (1.483 units, F-test = 217.726, *p* < 0.001) reported a meaningful improvement. Post-COVID-19 pneumonia patients presented malnutrition and functional impairment. An interdisciplinary rehabilitation program, including personalized nutritional intervention, was effective for post-hospital COVID-19 pneumonia nutritional/functional rehabilitation.

## 1. Introduction

COVID-19 pneumonia is an infectious disease that primarily compromises the respiratory tract but may evolve into multi-organ failure and be fatal [1,2,3,4,5]. Although the general population is susceptible to COVID-19 pneumonia, most hospitalized patients are older adults and/or with nutrition-related chronic diseases (obesity, diabetes, hypertension, and cardiovascular diseases) [6]. Several factors have been associated with an increased risk of developing severe COVID-19 pneumonia and worse prognosis, such as hypoalbuminemia, lymphopenia, sarcopenia/fragility, high body mass index (BMI), and obesity, which are also related to malnutrition [7]. Due to complications of the disease, many patients are hospitalized in intensive care units (ICU) to have proper medical support. For those patients and particularly in long-term care, ICU stay is a well-known cause of malnutrition, with skeletal muscle mass loss and dysfunction, leading, after ICU discharge, to poor quality of life, disability, and morbidities such as increased inflammation, gastrointestinal alterations, loss of taste and smell [8]. This situation is much more frequent in older adults with nutrition-related disorders that increase the risk of severe COVID-19 pneumonia and increased risk of death during hospitalization [9,10,11,12,13].

Proper nutrition and good nutritional status have gained relevance in the context of the SARS-CoV-2 pandemic. Food and nutrition play a crucial role in preventing severe disease and are becoming one of the priority areas of intervention to minimize the consequences of this viral infection [2]. Having an adequate diet contributes to a better nutritional status and an effective immune response, and it decreases the risk of severe evolution and complications [2]. Therefore, after discharge from the hospital, many patients that recovered from COVID-19 pneumonia undergo specialized rehabilitation in an assisted recovery care facility. The present study aimed to assess the effectiveness of an intensive rehabilitation program for the nutritional and functional status improvement of patients after COVID-19 pneumonia, admitted into four assisted recovery facilities.

## 2. Materials and Methods

This study was conducted between 1 February and 4 June of 2021 (the finish line of COVID-19 3rd wave in Portugal).

We included in the study patients hospitalized in four specialized rehabilitation care units (assisted recovery care facilities of the same Portuguese health group located in different districts) who had a history of COVID-19 pneumonia and were willing to participate in the study after invitation, signing the informed consent. Each patient was followed for 30 days, as an inpatient in the rehabilitation care units from admission (T0) until discharge (T2). They were submitted to an interdisciplinary rehabilitation program including 30 days of intensive rehabilitation, physiotherapy, occupational therapy, psychological intervention, speech therapy, and personalized nutrition care, associated with medical and nursing care. The nutritional care program for each patient was individualized based on the European Society for Clinical Nutrition and Metabolism recommendations in the “ESPEN expert statements and practical guidance for the nutritional management of individuals with SARS-CoV-2 infection” document [14].

Each patient was evaluated at three different moments: (T0) Upon admission to the unit; (T1) After 15 days of the intervention program; (T2) Within 48 h before discharge, after 30 days of the intervention program.

Each assessment included the following parametersThe anthropometric data evaluation was carried out according to the procedures described in the manual of the International Society for the Advancement of Kinanthropometry (ISAK) [15]. Measurements were obtained by the same instruments and researcher to minimize interobserver variability. In every one of the three different moments, each parameter was measured three times and each value represents the mean individual value:1.1.Body mass index (BMI) was valued according to age [16,17]. BMI was obtained in most patients using the equation Weight (Kg)/Height (m)^2^. If patients were bedridden and could not stand up for weight and height evaluation, BMI was estimated using the mid-upper arm circumference (MUAC) and regression equations described by Powell-Tuck and Hennessy [18]. It has already been demonstrated that this methodology accurately estimates BMI in malnourished patients. [19].1.2.Mid Upper Arm Circumference (MUAC) was evaluated using an inextensible measuring tape, with a 1 mm resolution. MUAC results from evaluating several tissues representing fat and lean mass.1.3.Tricipital skinfold (TSF), was measured using a Lange Skinfold caliper with a 1 mm resolution. TSF evaluates the subcutaneous adipose tissue and estimates adipose reserves.1.4.The Mid-Arm Muscle Circumference (MAMC) was calculated according to the equation: MAMC = MUAC (cm) − 0.314 × TSF (mm). The MAMC allows us to estimate lean and muscle mass.

For each patient, MUAC, MAMC, and TSF were compared with reference values of the National Health and Nutrition Examination Survey (NHANES), through the comparison with the Frisancho reference tables [20]. The adequacy of those indexes was determined through the following equation:

Index adequacy (%) = index obtained/index 50 percentile × 100. To achieve the index’s nutritional status adequacy, the results were classified according to the Blackburn and Thornton criteria [21].

Although nutritional evaluation could benefit from sophisticated devices for measuring body composition, such as bioelectrical impedance analysis (BIA) or CT Scan analysis, those devices were unavailable in all the centers included in this multicentric study. Although less precise, BMI and anthropometry are inexpensive and very widespread nutritional evaluation tools, classically used as an approach to the evaluation of fat/lean mass [22] and available everywhere, even in institutions with scarce resources.

2.Functional assessmentHandgrip strength (HGS) was assessed with a calibrated pressure dynamometry in Kg resolution (JAMAR^®^ model:5032P). For HGS measurement, the patients looked straight and were comfortably seated. The arms were slightly spread out so that the hands did not touch the thighs, and the elbows were extended. When the examiner provided the start signal, the individual gripped as firmly as possible while maintaining the specified posture. The grip strength was measured twice for both hands, and each hand’s maximum value was recorded. The average of the maximum values for both hands was used for the analysis in the three different evaluation moments.3.Nutritional assessment questionnaireThe Portuguese version of the validated questionnaire “Mini Nutritional Assessment” (MNA®) was used to obtain a nutritional assessment of each patient at admission and 48 h before discharge [23]. The MNA^®^ form was provided by Société des Produits Nestlé SA 1994, Revision 2009, Vevey, Switzerland, Trademark Owners, which holds the copyright of the instrument: http://www.mna-elderly.com/ (accessed on 16 January 2021).4.Personalized nutritional care programFor the patients that underwent this rehabilitation program, the energy requirements were calculated at 30 kcal/kg body weight/day. Concerning protein requirements, the objective was to achieve a minimum of 1 g protein/kg body weight/day. Nevertheless, these values were individually adjusted regarding nutritional status, physical activity level, disease status, and tolerance. All patients received dietary counseling, and when necessary, their food was fortified with modular protein. When needed to achieve the nutritional goals, high-caloric and high-protein oral nutritional supplements (ONS) were used according to personalized options.

### 2.1. Intensive Care Unit Stay, Sex, and Nutritional/Functional Evolution

The nutritional and functional evolution tool’s results were evaluated on all the patients and also according to sex and need for ICU care.

### 2.2. Statistical Analysis

The descriptive analysis results and plots were obtained using MS Excel while statistical tests and models were performed by SPSS software version 26. Models were either fitted including all subjects’ data or limited to care unit type (ICU/non-ICU) groups.

Initially, we fitted two-level hierarchic models with the rehabilitation care units district as random effects two-level variable. Still, due to the district component variance being redundant for all outcomes, fitting changed to one-level models.

Anthropometric indexes, MNA^®^, and HGS data collected at baseline (T0) and intervention periods (T1 and T2) were used to calculate the difference between T0 and T2 variables (intervention effect), with T0, T1 and T2 as repeated-measures-dependent variables.

The intervention effect variables were used to fit generalized linear models (GLM) and the repeated-measures-dependent variables were used to fit generalized estimating equations (GEE) models. Both models type accommodate categorical (MNA^®^, TSF, MAMC and MUAC) or quantitative (BMI and HGS) dependent variables. GLM and GEE models followed model-based estimator covariance matrix and parameter estimation by maximum likelihood method with type III analysis of model effects.

The categorical dependent variable MNA^®^ included the levels “Normal nutritional status”, “Risk of malnutrition” and “Malnutrition”, and the variables TSF, MAMC and MUAC included the levels “low “and “normal”. The first level of each variable was used as reference in the comparisons. The categorical outcomes were fitted under a logistic regression with multinomial (ordinal) distribution and cumulative logit as the link function, while quantitative outcomes were fitted under linear models type.

GEE models were tested for within subjects’ trends through F-test for polynomial components, followed by marginal means pairwise comparisons for every trend found. To evaluate the effect of the intervention program on the outcome across the study timeframe (T0–T2 interval), we included an intervention period time variable in the model and tested the associated coefficient with the Wald Chi-Square test.

Furthermore, models included patients’ age and gender as covariates to evaluate if the intervention outcome was the same in the different patient age strata and for male and female patients.

All statistical tests were considered significant if the associated *p*-value were <0.05.

## 3. Results

### 3.1. Subjects

We recruited 125 patients convalescing from COVID-19 pneumonia. Of these patients, 7 were excluded from the analysis due to not finishing the protocol (5 died during the rehabilitation stay, and the other 2 saw their clinical condition worsen and returned to the hospital). Of the 118 patients who had completed the personalized nutritional program intervention, 58 were females and 60 were males. Ages ranged from 41 to 90 years (mean: 71.9 years; median: 72 years): 27 patients were aged less than 65 years old, and 91 were 65 years old or older (Table 1).

### 3.2. Patients’ ICU/Non-ICU Status

From a total of 118 patients convalescing from COVID-19 pneumonia, 57 had an intensive care unit (ICU) provenience and 61 did not have the necessity for ICU internment. In the ICU group of patients, 29 were females and 28 were males. Ages ranged from 44 to 88 years (mean: 71.4 years; median: 71 years), and 13 patients were younger than 65 years old. In the non-ICU group of patients, 29 were females and 32 were males. Ages ranged from 41 to 90 years (mean: 72.6 years; median: 74 years), and only 14 patients were younger than 65 years old (Table 1).

### 3.3. Body Mass Index

BMI was obtained in all assessment moments (T0; T1; T2) for 118 patients. For 12 patients, BMI was estimated using the Powell-Tuck and Hennessy regression equations according to our previous experience [17]. The WHO BMI classification is used according to age [16,17] and evaluation results are reported in Table 1.

BMI has, on average, increased significantly, by 0.436 units in T0–T1 (*p* < 0.001) and by 0.206 units in T1–T2 (*p* = 0.007). Moreover, BMI showed a significant positive linear trend over time (0.642 units, F-test = 26.458, *p* < 0.001) with a slight quadratic slope decrease towards the last intervention period (T1–T2; F-test = 3.735, *p* = 0.056). In the first 15 days, BMI increased twice as much as in the last 15 days.

When the population was split into ICU and non-ICU patients, the evolution of BMI was slightly different. In ICU patients, the evolution of BMI showed a significant quadratic trend (F-test = 6.787, *p* = 0.012) suggesting a faster improvement when compared to the non-ICU population, where BMI dynamics showed a significant positive linear trend over time (F-test = 7.946, *p* = 0.007), but not quadratic (F-test = 0.568, *p* = 0.454).

### 3.4. Mid Upper Arm Circumference (T0; T1; T2)

The MUAC was evaluated in all patients (Table 2). MUAC does not show a significant positive trend during the intervention period (T0–T2) (0.322 units, *p* = 0.474). Regarding the evaluation moments T0–T1 and T1–T2, MUAC increases in both periods, but not significantly.

Frisancho grading results (Table 2) in ICU patients have shown a global increase trend (*p* = 0.025), especially between T0 and T2 (*p* = 0.007) when comparing the first (T0; T1) and second (T1; T2) fortnight periods. MUAC measurements in non-ICU have not significantly changed (*p* = 0.348) along the intervention period (T0–T2).

### 3.5. MUAC Adequacy (%) (T0; T1; T2)

MUAC adequacy (%) was calculated in all subjects (Table 2) and did not show a meaningful change over time for the T0–T2 time frame (*p* = 0.398), even after adjusting for ICU and non-ICU patients (*p* = 0.929 and *p* = 0.856, respectively).

### 3.6. Tricipital Skinfold (T0; T1; T2)

The TSF was completed in all 118 patients (Table 3), in ICU and non-ICU decreased during the intervention period without a statistical difference (*p* = 0.929 and *p* = 0.358, respectively).

### 3.7. TSF Adequacy (%) (T0; T1; T2)

TSF adequacy (%) (Table 3) does not show a significant change over time (T0–T2) (*p* = 0.284), and even when divided into ICU and non-ICU patients also do not show a significant change along the entire intervention period (T0–T2) (*p* = 0.426 and *p* = 0.558, respectively).

### 3.8. Mid-Arm Muscle Circumference (T0; T1; T2)

The MAMC was completed in all 118 patients (Table 4) and showed an increasing trend over time, but with no statistical relevance (T0–T2) (*p* < 0.299).

The obtained Frisancho grading results (Table 4), in ICU and non-ICU showed an increasing trend towards the intervention final intervention period (T0–T2), although with no statistical relevance (*p* = 0.165 and *p* = 0.513, respectively).

### 3.9. MAMC Adequacy (%) T0; T1; T2

MAMC adequacy (%) (Table 4) showed a trend for positive effect over time (T0–T2) but with no statistical support. MAMC adequacy (%) in ICU and non-ICU showed a positive effect trend over time (T0–T2) (*p* = 0.007 and *p* < 0.001, respectively) especially in the first 15th days (*p* < 0.000; *p* = 0.021).

### 3.10. Hand Grip Strength (T0; T1; T2)

The HGS was completed in all patients (Table 5). F-test statistics showed that HGS increased linearly over the intervention periods (4.131 units, F-test = 82.540, *p* < 0.001), with a slight decrease in the last measurement. Furthermore, pairwise comparisons showed that T1 and T2 intervention periods had meaningful increases regarding the precedent one (2.554 units, *p* < 0.01).

Regarding HGS, we observed a linear increase over intervention periods between the three assessment moments in ICU (4.035 units, F-test = 48.215, *p* < 0.001) and non-ICU (4.185 units, F-test = 37.023, *p* < 0.001) patients.

### 3.11. Mini Nutritional Assessment at T0 and T2

The MNA^®^ was completed for all patients at the admission (T0) and 48 h before discharge (T2) (Table 6).

The timeframe between T0 and T2 had a significant effect on the global MNA^®^ score nutritional status (1.483 units, F-test = 217.726, *p* < 0.001) and in ICU and non-ICU groups (*p* < 0.001 for both).

### 3.12. Sex Effect

The covariate sex had no significant effect on the outcome of the tools for all the sample size (*p* = 0.919), and within the ICU and non-ICU subsets (*p* = 0.574 and *p* = 0.530, respectively).

### 3.13. Age Effect

In general, higher age was associated with increased BMI, MUAC, and MAMC recovery, probably because of lower values at admission. The age effect within ICU and non-ICU groups for BMI, MUAC, and MAMC recovery, was similar to the one sourced from the global sample. Regarding MNA^®^ recovery, the only significant age effect detected was in the ICU patients’ group (*p* = 0.009), probably due to including, on average, older patients than the non-ICU patients’ group.

## 4. Discussion

We assessed the nutritional and functional status of people recovering from COVID-19 pneumonia, before and after an interdisciplinary rehabilitation program. Our study is comparable to others that use nutritional intervention for post-COVID-19 pneumonia patients’ rehabilitation [24,25]. However, when compared with other studies performed in a hospital context, the present study was conducted in a post-hospital rehabilitation setting with a large number of patients, albeit with a single-month intervention period [26]. Our results showed that many patients display a poor nutritional and functional status when they are received at rehabilitation units, having been transferred from hospitals after treatment for COVID-19 pneumonia.

The nutritional and functional status of these patients significantly improved at the end of the rehabilitation program (30 days), although an important number of patients still presented several clinical variables with values below the reference values for normality.

Importantly, we saw that the evolution of the nutritional and functional status of the patients that underwent the interdisciplinary rehabilitation program was positive. Almost all the analyzed nutritional and functional variables had improved. However, only BMI, HGS, and MNA^®^ presented a significant change, in this timeframe. On the other hand, global changes in MUAC/MUAC Adequacy, TSF/TSF Adequacy, and MAMC/MAMC Adequacy were positive but not significant over this intervention program.

BMI reported a significant increase between the time of admission and 30 days after the nutritional intervention, suggesting an improvement in fat and lean mass. Nevertheless, the contrast of the TSF results with the MAMC results, when adjusted to age, indicates that the fat-free tissue is more depleted than the fat tissue during the COVID-19 pneumonia disease process. In fact, TSF is normal/high in 85.5% of the patients since baseline, proving that, in this clinical setting, disease-associated malnutrition impacts mostly lean tissue, with a major functional impact [3]. The difficulty in measuring this parameter in bedridden or debilitated patients, such as COVID-19 pneumonia patients, may be the reason for the lack of statistical significance of the TSF improvement observed.

We saw that changes in BMI by class (low, normal, high) occur between the normal and high groups. Some patients went from normal to high classification. However, there were no changes from low to normal class. Within 1 month of intervention, it was not possible to change the BMI profile of individuals with low to normal BMI. These results suggest that underweight patients needed more time for rehabilitation to recover, because their baseline status was probably worse, and possibly they may have lost more lean mass than patients with normal or high BMI. With the personalized nutritional intervention, seven patients were no longer classified as malnourished and three as overweight, to be classified as eutrophic when assessed by MUAC adequacy (%), allowing 8.5% of patients to acquire a normal weight classification. This result suggests an improvement in general nutrition status since this parameter evaluates both fat and lean mass. In fact, MAMC/MAMC Adequacy presents a slight increase between T0 and T2, 30 days after the beginning of the personalized intervention, which suggests an improvement in muscle mass of those patients. In MAMC, we noticed a superior increase in T1–T2, suggesting a meaningful increase in lean mass. Still, an extended rehabilitation period might be necessary, as we observed in other studies [25]. HGS significantly increased between T0 and T2, reflecting an improvement in functional status. Altogether, it seems that muscle mass and muscle function were mostly affected by COVID-19 pneumonia, and display the most impressive improvement due to the interdisciplinary rehabilitation program with personalized nutrition.

The present study used the MNA^®^ as a global nutritional assessment tool for all the patients. This tool is validated and used primarily for people 65 years old or older, but for better consistency, our study included a few patients younger than 65. The MNA^®^ results presented a consistent mean increase between T0 and T2, which was statistically significant. These results also represent a positive patient response to this personalized nutritional intervention program, decreasing the risk of malnutrition, when evaluated with one of the most reliable validated nutritional assessment tools in clinical practice [27].

The improvements were particularly remarkable for most of the variables analyzed during the last 15 days. These improvements suggest that in the post-hospital stay, as soon as personalized nutrition interventions begin, patients present a positive response, but for complete nutritional rehabilitation, patients need more intervention time, or at least some patients could benefit from a more extended rehabilitation period than 30 days.

It is known that COVID-19 pneumonia has more severe manifestations in older adults, leading to high functional and nutritional losses [28]. We have assessed the data against the age factor and the results suggest that the older adults possibly had a higher nutritional and functional loss during the period of the disease, which enabled them to have a greater margin of recovery.

Globally, this interdisciplinary rehabilitation with a personalized nutrition program was effective for recovery after hospital discharge from COVID-19 pneumonia, in both nutritional and functional status.

### ICU vs. Non-ICU

When subjects are classified according to the severity of their disease (with previous ICU internment or not), the generality of anthropometric parameters was more preserved at baseline in ICU patients than in non-ICU patients. The authors believe that despite the severity of the disease, the internment in the ICU may have benefited from greater attention to nutritional care. However, HGS was more reduced in these patients, reflecting a functional deterioration caused by a longer period of immobilization.

Looking into the evolution during the rehabilitation program, our results showed a significant evolution along the intervention period in ICU patients in BMI, MUAC, MAMC Adequacy, HGS, and MNA^®^. These results possibly suggest, in ICU patients, a bigger depletion in fat and muscle mass compared to their pre-disease nutritional status, and therefore more potential for improvement.

In this study, we have some limitations. Our sample is a convenience sample, with a limited number of patients. Furthermore, given the lack of a control group, it is not possible to say that the recovery is only the effect of the protocol applied in our study. Additionally, ICU stay was considered a marker of severe disease, but the authors did not have access to the length of the ICU period. Another limitation is the rehabilitation time. These patients have just a 30-day program in this institution, because that was a public health protocol. Compared to other studies, it was a short period. It is probably that a longer intervention time could have led to better nutritional and functional outcomes, and for the authors, it is clear that at least some patients would benefit from a longer public health protocol. Another limitation of this study stems from its multicentric nature. The authors avoided sophisticated instruments and chose to use simple, widespread nutrition evaluation tools that could be used in every one of the four centers involved and that can be used in a wide range of other rehabilitation institutions.

## 5. Conclusions

After hospital discharge for COVID-19 pneumonia, patients presented malnutrition and functional impairment, evaluated using BMI, arm anthropometry, MNA^®^, and HGS. Arm anthropometry demonstrated an important lean/fat-free mass depletion with a reduced impact on fat reserves evaluated with TSF. Recovery seemed to be more noticeable for fat-free mass, and functional HGS. The COVID-19 pneumonia patients who presented a better recovery were the older ones (included in the ICU group), probably because they presented a worse clinical status at the beginning of the rehabilitation program. In fact, in this cohort, the results of the program were more notable during the last two weeks, suggesting that some patients may require a longer intervention.

## 6. Implication for the Future

Globally, an interdisciplinary rehabilitation program, including personalized nutritional intervention, is effective for post-hospital COVID-19 pneumonia patients. Our study suggests the potential usefulness of applying for similar interdisciplinary programs with a personalized nutritional intervention in a wide range of clinical settings. The present personalized nutritional program should be considered a model for active and intense rehabilitation.

## Figures and Tables

**Table 1 nutrients-14-02501-t001:** Characterization of subjects and respective results from Body Mass Index (BMI) in all evaluation moments (T0; T1; T2).

		Global			ICU Group			Non-ICU Group		
		<65 year	≥65 year	Total	<65 year	≥65 year	Subtotal	<65 year	≥65 year	Subtotal
	n	27	91	118	13	44	57	14	47	61
Sex	Male	16	44	60	7	23	28	9	23	32
	Female	11	47	58	6	21	29	5	24	29
Age	Range	41/64	65/90	41/90	44/64	65/88	44/88	41/64	65/90	41/90
	Mean	56.6	76.6	71.9	57	75.66	71.4	56.1	77.5	72.6
	Median	58	76	72	58	74	71	58	77	74
BMI T0 Kg/m^2^	Range	15.67/ 46.87	13.11/ 46.93	13.11/ 46.93	17.30/46.87	13.11/ 38.04	13.11/ 46.87	15.68/ 39.78	13.56/ 46.93	13.11/ 46.87
	Mean	27.02	24.17	24.74	27.66	24.34	25.10	26.41	24.01	24.56
	Median	25.35	24.02	22.53	24.02	25.18	24.98	25.39	23.58	23.83
	Low (n/%)	2/7.4	32/35.1	34/28.8	1/7.6	15/34	6/10.5	1/7.1	18/38.3	7/11.5
	Normal (n/%)	11/40.7	34/37.3	45/38.1	6/46.2	14/32	23/40.3	5/35.7	19/40.4	31/50.8
	High (n/%)	14/51.8	25/27.4	39/33	6/46.2	15/34	28/49.2	8/57.2	10/21.3	23/37.7
BMI T1 Kg/m^2^	Range	15.67/ 46.87	13.46/ 47.80	13.46/ 47.80	17.67/46.87	13.46/ 38.44	13.46/ 47.80	15.68/ 38.75	16.26/ 47.80	15.67/ 47.80
	Mean	27.14	24.70	24.99	27.71	25.05	25.66	26.60	24.37	24.88
	Median	25.23	23.80	22.90	24.42	25.89	25.74	25.46	23.41	23.77
	Low (n/%)	2/7.4	32/35.1	34/28.8	1/7.6	14/31.8	5/8.7	1/7.1	18/38.3	4/6.5
	Normal (n/%)	11/40.7	31/34	42/35.5	6/46.2	11/25	22/38.6	2/14.3	19/40.4	33/54.1
	High (n/%)	14/51.8	28/30.7	42/35.5	6/46.2	19/43.2	30/52.7	11/78.6	10/21.3	24/39.4
BMI T2 Kg/m^2^	Range	15.68/ 46.87	13.81/ 47.80	13.82/ 47.80	17.63/ 46.87	13.87/ 38.67	13.81/ 46.87	15.68/ 38.20	16.03/ 47.80	15.68/ 47.80
	Mean	27.13	24.97	25.13	24.68	25.38	25.90	26.61	24.59	25.05
	Median	25.33	24.80	23.43	24.88	26.26	26.08	25.50	20.06	24.74
	Low (n/%)	2/7.4	32/35.1	34/28.8	1/7.6	15/34.1	4/7.1	1/7.1	17/36.2	5/8.1
	Normal (n/%)	10/37.1	29/31.8	39/33	6/46.2	9/20.5	23/40.4	4/28.6	20/42.6	27/44.3
	High (n/%)	15/55.5	30/32.9	45/38.1	6/46.2	20/45.4	30/52.5	9/64.3	10/21.2	29/47.6

ICU—intensive care unit; BMI—Body Mass Index; BMI classification according to age, <65 year low BMI is <18.5 Kg/m^2^, normal BMI is between 18.5 Kg/m^2^ and <25 Kg/m^2^, and high BMI is ≥25 Kg/m^2^ ≥65 year, low BMI is <22 Kg/m^2^, a normal BMI is between 22 Kg/m^2^ and <27 Kg/m^2^, and high BMI is ≥27 Kg/m^2^.

**Table 2 nutrients-14-02501-t002:** Characterization of subjects and respective results Mid Upper Arm Circumference and from Mid Upper Arm Circumference Adequacy (%) in all evaluation moments (T0; T1; T2).

		Total	ICU Group	Non-ICU Group
MUAC T0 (cm)	Mean	26.08	26.40	25.72
	Median	25.5	24	25
	Low (n/%)	76/64.4	36/63.2	40/65.6
	Normal (n/%)	42/35.6	21/36.8	21/34.4
MUAC T1 (cm)	Mean	26.10	26.84	25.39
	Median	26	26	26
	Low (n/%)	74/62.70	32/56.10	42/68.90
	Normal (n/%)	44/37.30	25/43.90	19/31.10
MUAC T2 (cm)	Mean	26.40	27.06	25.75
	Median	26	26	26
	Low (n/%)	66/55.90	29/50.90	37/60.70
	Normal (n/%)	52/44.10	28/49.10	24/39.30
MUAC Adequacy(%) T0	Malnutrition (n/%)	74/62.70	34/59.60	40/65.60
	Eutrophic (n/%)	35/29.70	18/31.60	17/27.90
	Overweight (n/%)	9/7.60	5/8.80	4/6.50
MUAC Adequacy(%) T1	Malnutrition (n/%)	68/57.60	31/54.40	37/60.70
	Eutrophic (n/%)	44/37.30	22/38.60	22/36.10
	Overweight (n/%)	6/5.10	4/7.00	2/3.20
MUAC Adequacy(%) T2	Malnutrition (n/%)	67/56.80	31/54.40	36/59.00
	Eutrophic (n/%)	45/38.10	23/40.40	22/36.10
	Overweight (n/%)	6/5.10	3/5.30	3/4.90

ICU—intensive care unit; MUAC—Mid Upper Arm Circumference; MUAC Adequacy (%)—Mid Upper Arm Circumference Adequacy (%).

**Table 3 nutrients-14-02501-t003:** Characterization of subjects and respective results from Tricipital Skinfold (TSF) and Tricipital Skinfold Adequacy (%) (TSF Adequacy (%)) in all evaluation moments (T0; T1; T2).

		Total	ICU Group	Non-ICU Group
TSF T0 (mm)	Mean	16.94	15.60	18.19
	Median	14	16	18
	Low (n/%)	17/14.4	10/17.5	7/11.5
	Normal (n/%)	101/85.5	47/82.5	54/88.5
TSF T1 (mm)	Mean	16.51	15.43	17.47
	Median	14	12	18
	Low (n/%)	16/13.5	9/15.7	7/11.5
	Normal (n/%)	102/86.4	48/84.3	54/88.5
TSF T2 (mm)	Mean	16.37	15.53	17.09
	Median	14	12	16
	Low (n/%)	11/9.3	6/10.5	5/8.2
	Normal (n/%)	107/90.6	51/89.5	56/91.8
TSF Adequacy(%) T0	Malnutrition (n/%)	43/36.4	24/42.1	19/31.1
	Eutrophic (n/%)	19/16.1	13/22.8	6/9.8
	Overweight (n/%)	56/47.5	20/35.1	36/59.0
TSF Adequacy(%) T1	Malnutrition (n/%)	44/37.3	22/38.6	22/36.1
	Eutrophic (n/%)	20/16.9	12/21.0	3/4.9
	Overweight (n/%)	54/45.8	23/40.4	36/59.0
TSF Adequacy(%) T2	Malnutrition (n/%)	42/35.6	23/40.4	19/36.1
	Eutrophic (n/%)	22/18.6	11/19.2	11/18.0
	Overweight (n/%)	54/45.8	23/40.4	31/50.8

ICU—intensive care unit; TSF—Tricipital Skinfold; TSF Adequacy (%)—Tricipital Skinfold Adequacy (%).

**Table 4 nutrients-14-02501-t004:** Characterization of subjects and respective results from Mid-Arm Muscle Circumference (MAMC) and Mid-Arm Muscle Circumference Adequacy (%) (MAMC Adequacy (%)) in all evaluation moments (T0; T1; T2).

		Total	ICU Group	Non-ICU Group
MAMC T0 (mm)	Mean	20.90	21.55	20.01
	Median	20.76	21.23	20.60
	Low (n/%)	114/96.6	55/96.5	59/96.7
	Normal (n/%)	4/3.4	2/3.5	2/3.3
MAMC T1 (mm)	Mean	20.90	22.01	19.90
	Median	20.76	22.23	19.09
	Low (n/%)	114/96.6	55/96.5	59/96.7
	Normal (n/%)	4/3.4	2/3.5	2/3.3
MAMC T2 (mm)	Mean	21.20	22.19	20.38
	Median	21.26	22.23	20.86
	Low (n/%)	109/92.3	52/91.2	57/93.4
	Normal (n/%)	9/7.6	5/8.8	4/6.6
MAMC Adequacy(%) T0	Malnutrition (n/%)	104/88.1	48/84.2	56/91.8
	Eutrophic (n/%)	12/10.2	7/12.3	5/8.2
	Overweight (n/%)	2/1.7	2/3.5	0/0
MAMC Adequacy(%) T1	Malnutrition (n/%)	105/89.0	52/91.2	53/86.9
	Eutrophic (n/%)	12/10.2	4/7.0	8/13.1
	Overweight (n/%)	1/0.8	1/1.8	0/0
MAMC Adequacy(%) T2	Malnutrition (n/%)	103/87.3	49/86.0	54/88.5
	Eutrophic (n/%)	14/11.9	7/12.3	7/11.5
	Overweight (n/%)	1/0.8	1/1.8	0/0

ICU—intensive care unit; MAMC—Mid-Arm Muscle Circumference; MAMC Adequacy (%)—Mid-Arm Muscle Circumference Adequacy (%).

**Table 5 nutrients-14-02501-t005:** Characterization of subjects and Hand Grip Strength (HGS) results from in all evaluation moments (T0; T1; T2).

		Total	ICU Group	Non-ICU Group
HGS T0 (kg)	Mean	13.64	13.17	13.49
	Median	12	12	12
HGS T1 (kg)	Mean	15.66	15.72	16.09
	Median	14	16	16
HGS T2 (kg)	Mean	17.18	17.24	17.67
	Median	16	18	18

ICU—intensive care unit; HGS—Hand Grip Strength.

**Table 6 nutrients-14-02501-t006:** Characterization of subjects and respective results from Mini Nutritional Assessment (MNA^®^) (T0; T2).

		Total	ICU Group	Non-ICU Group
MNA^®^ T0	Malnutrition (n/%)	14/11.2	9/15.8	5/8.2
	Risk of malnutrition (n/%)	69/58.4	33/57.9	36/59.0
	Normal nutritional status (n/%)	35/29.6	15/26.3	20/32.8
MNA^®^ T2	Malnutrition (n/%)	10/8.5	7/12.2	3/4.9
	Risk of malnutrition (n/%)	72/61.0	34/59.6	38/62.3
	Normal nutritional status (n/%)	36/35.5	16/28.2	20/32.8

ICU—intensive care unit; MNA^®^ classification score < 17 indicates malnutrition; score 17–23.5 indicates risk of malnutrition, and score > 24 indicates normal nutritional status.

## Data Availability

The data presented in this study are available on request from the first author.

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
