# Peer review of "The Effects of an Intensive Rehabilitation Program on the Nutritional and Functional Status of Post-COVID-19 Pneumonia Patients"

_nutrients, 2022, doi:10.3390/nu14122501_

Round 1

Reviewer 1 Report

  1. Nutritional status in nowdays and with the availability of devices for measuring body composition cannot be assessed on the basis of BMI and anthropometry. Hydration status, edema, poor reliability ... all this affects the ASSESSMENT of body composition with anthropometric methods.
  2. MNA is a tool for assessing the risk of malnutrition. If it is positive, it does not mean that the person is malnourished. I miss at least the use of GLIM criteria for malnutrition.
  3. Based on the results of research in the field of rehabilitation after severe form of COVID-19 (ICU), it is clear that rehabilitation is long-term. 30 days seems too little. It is also not known when participants overcame COVID-19; how long they have been in the ICU ... Duration is an important component.
  4. 3D bar charts are misleading; showing the same results with a table and graph is pointless.

Author Response

Dear Reviewer, 

we appreciate your recommendations. Please see the attachment. 

Best Regards 

Reviewer 2 Report

In this article, the authors evaluated the effectiveness of an intensive rehabilitation program on the nutritional/functional status of patients recovering from COVID-19 pneumonia. It is a nice manuscript with an important clinical meaning. However, several points need to be addressed for full consideration. English editing by a native speaker is needed. Please avoid the term elderly/elders since it is pejorative. It would be better older people or older adults.

Abstract

Page 1; line 14: Please change “elders” to “older adults”

Page 1; line 15: Please change “chronic diseases” to “issues”

Page 1; line 16: Please change “muscle mass loss” to “muscle wasting”

Introduction

This section should better characterize the effect of COVID-19 infection (i.e., increased inflammation, gastrointestinal alterations, loss of taste and smell) on nutritional status and vice versa (doi.org/10.3390/nu13051616; doi: 10.1002/jcsm.12589; doi.org/10.1038/s41430-020-00795-0)

Page 1; line 39: Please change “elderly” to “older adults”

Page 1; line 39: However, the assessment of nutritional status via BMI presents several limitations, especially in older people (doi.org/10.3389/fendo.2020.581356)

Materials and methods:

I suggest reorganizing this section with different subheadings (i.e., 2.1 personalized nutritional care program, 2.2 anthropometric measures)

Page 2; line 62: Please change “occurred” to “was conducted”

Page 2; line 64: “hospitalized in four specialized rehabilitation care units”. Please specify if these four units refer to the same hospital or to four different centers.

Discussion

The limitations of the study are missing (i.e., the small sample size, etc.)

Author Response

(The authors gave the same response as above.)

Reviewer 3 Report

I congratulate the authors on the work done on this relevant subject. I would like to contribute with some comments, suggestions and questions to the authors. 

Firstly, the document had formatted carelessly and should be better formatted e.g. abstract, indents, bullets, table etc.

Introduction:

  • Line 49 I suggest to mention that the malnutrition status also correlates with an increased risk of death during hospitalisation.
  • A good references are:

    Miguélez M, Velasco C, Camblor M, Cedeño J, Serrano C, Bretón I, Arhip L, Motilla M, Carrascal ML, Morales A, Brox N, Cuerda C. Nutritional management and clinical outcome of critically ill patients with COVID-19: A retrospective study in a tertiary hospital. Clin Nutr. 2021 Nov 1:S0261-5614(21)00499-4. doi: 10.1016/j.clnu.2021.10.020.

    Czapla M, Juárez-Vela R, Gea-Caballero V, ZieliÅ„ski S, ZieliÅ„ska M. The Association between Nutritional Status and In-Hospital Mortality of COVID-19 in Critically-Ill Patients in the ICU. Nutrients. 2021; 13(10):3302. https://doi.org/10.3390/nu13103302

Methodology:

  • The description and application of the measurement methods are appropriate but I think that you should add in methodology information that MNA form provided by Société des Produits Nestlé SA 1994, Revision 2009, Vevy, Switzerland, Trademark Owners, which holds the copyright of the instrument: http://www.mna-elderly.com/) – in this website describes all rules how to citate this form.
  • Methodology section should be better organized. E.g. study design, study population etc.
  • This section should be completely rewriting - must be arranged correctly.
  • Result: Generally the analysis of the results was carried out in detail. In tabl1 In my opinion is better to use words “Survivors” instead “alive”

Result

  • My main comment is regarding the number of tables and figures. I think the use of 9 figures and 6 tables is too much, it dilutes your key findings/message. It might be beneficial to the reader to focus on fewer tables (perhaps 5 or less) and figures (perhaps 4 or less) . You could look to combine/condense tables together, or add them as a supplementary file.

  • Line 286 “The covariate sex had no significant effects on the trends of any variable even when 288 we separate the patients in ICU and non-ICU groups.” What’s does it mean? Please add P-value

Discussion

  • The discussion was conducted in an appropriate manner.
  • The study doesn’t have any limitation? Please add study limitation section on the end of discussion.

Conclusion

  • Generally The final conclusions are correct but I recommended to separate for clinical implication and conclusion. You have too much information in conclusion section. One part you should describe as implication.

Author Response

(The authors gave the same response as above.)

Round 2

Reviewer 3 Report

Thank you. Now is all right.

Author Response

we really appreciate your comments